# HIV-1 Integrase Strand Transfer Inhibitors and Neurodevelopment

**DOI:** 10.3390/ph15121533

**Published:** 2022-12-09

**Authors:** Emma G. Foster, Howard E. Gendelman, Aditya N. Bade

**Affiliations:** 1Department of Pharmacology and Experimental Neuroscience, University of Nebraska Medical Center, Omaha, NE 68198, USA; 2Department of Pharmaceutical Sciences, University of Nebraska Medical Center, Omaha, NE 68198, USA

**Keywords:** HIV-1, pregnancy, antiretroviral drugs, integrase strand transfer inhibitor, neurodevelopment, adverse events

## Abstract

Children born to mothers, with or at risk, of human immunodeficiency virus type-1 (HIV-1) infection are on the rise due to affordable access of antiretroviral therapy (ART) to pregnant women or those of childbearing age. Each year, up to 1.3 million HIV-1-infected women on ART have given birth with recorded mother-to-child HIV-1 transmission rates of less than 1%. Despite this benefit, the outcomes of children exposed to antiretroviral drugs during pregnancy, especially pre- and post- natal neurodevelopment remain incompletely understood. This is due, in part, to the fact that pregnant women are underrepresented in clinical trials. This is underscored by any potential risks of neural tube defects (NTDs) linked, in measure, to periconceptional usage of dolutegravir (DTG). A potential association between DTG and NTDs was first described in Botswana in 2018. Incidence studies of neurodevelopmental outcomes associated with DTG, and other integrase strand transfer inhibitors (INSTIs) are limited as widespread use of INSTIs has begun only recently in pregnant women. Therefore, any associations between INSTI use during pregnancy, and neurodevelopmental abnormalities remain to be explored. Herein, United States Food and Drug Administration approved ARVs and their use during pregnancy are discussed. We provide updates on INSTI pharmacokinetics and adverse events during pregnancy together with underlying mechanisms which could affect fetal neurodevelopment. Overall, this review seeks to educate both clinical and basic scientists on potential consequences of INSTIs on fetal outcomes as a foundation for future scientific investigations.

## 1. Introduction

Remarkable progress was achieved in the treatment and prevention of human immunodeficiency virus type-1 (HIV-1) infections through the development of and wide-spread use of antiretroviral therapy (ART). To date, approximately thirty antiretroviral drugs (ARVs) from eight classes are currently approved for treatment [1,2]. ART regimens have progressed from multiple pills of different ARVs taken several times a day to a daily single pill containing up to three ARVs [1,2]. Moreover, ART regimens are often modified following viral resistance, tolerability, and adverse events. Thus, when adhered to ART regimens, infected people can expect an extended quality of life. Nonetheless and despite such advances, seen over the past 40 years, HIV-1-infected pregnant women and their fetuses, in particular, remain vulnerable to adverse ARV effects. This is underscored by the potential risk of neural tube defects (NTDs) associated with periconceptional usage of dolutegravir (DTG) in Botswana, first recorded during 2018 [3].

The World Health Organization (WHO) and the Department of Health and Human Services (DHHS) recommend antiretroviral therapy (ART) be administered to all pregnant and breastfeeding women living with HIV-1 [4,5]. These established guidelines were implemented due to the critical role of ARVs in reducing vertical transmission of HIV-1 from mother to child and with the goal of improving maternal health. Indeed, in the pre-ART era, up to 40% of infants became HIV-1 infected from a mother with known infection. ART has significantly reduced the rate of vertical transmission to less than 1% in North America and European countries [6,7,8,9]. Currently recorded transmission rate is up to 2–28% in 21 focus countries in sub-Saharan Africa (UNAIDS epidemiological estimates 2021) [10]. However, along with such benefits of ART, a risk of adverse events following fetal exposure to ARVs was demonstrated, which remains a concern with the growing availability of ARVs, especially in resource-limited countries (RLCs). Indeed, over a million ARV-exposed HIV-1 uninfected children are born each year [11,12] (Figure 1). It is reported that 85% of HIV-1-infected pregnant women, worldwide, had access to ART in 2020 and the number of infected women conceiving while on ART has steadily increased [13]. For example, in the United Kingdom (UK), the proportion of infected pregnant women increased from 20% in 2000–2004 to 76% in 2015–2019 [9]. Thus, in proportion the numbers of in utero ART exposed children will also increase in coming years due to better recordings of HIV-1 infection and ARVs availability in RLCs.

In recent years, integrase strand transfer inhibitor (INSTI) use, in particular, has become widespread. INSTIs are part of preferred first- and second- line ART regimens [1,2]. This is due to high potency and inherent barriers to drug resistance [14]. In particular, availability of DTG-based regimen substantively increased in the past 5 years. A breakthrough pricing agreement made in September 2017 accelerated the availability of generic DTG-based regimens in RLCs at the cost of $75 US dollars/person/year. Up to one hundred RLCs have implemented transitioning to DTG-based regimen by mid-2020 [15]. Indeed in 2019, 6.9 million people had access to generic DTG-based regimen and up to 15 million people worldwide will be treated with this DTG-based regimen by the year 2025 [15,16,17,18]. This includes women of child-bearing age who remain a significant infected population (UNAIDS data, 2021) [13]. It is estimated that there are around 15.5 million women, 15–49 years old, living with HIV-1 worldwide. In 2020, women accounted for about 50% of all new HIV-1 infections and 63% in sub-Saharan Africa [13]. Moreover, increasing pretreatment resistance to non-nucleoside reverse transcriptase inhibitors (NNRTIs) in RLCs, especially in women, increases the demand for the usage of DTG-based regimens [5,19]. In total, due to current inclusion of DTG-based regimen in clinical guidelines for treatment of pregnant women irrespective of trimester or those of child-bearing potential [4,5], and rising pretreatment NNRTI resistance in RLCs, the majority of HIV-1 infected women are being treated with DTG-based regimens globally. Apart from DTG, raltegravir (RAL) is also recommended as a preferred INSTI for use in ARV-naïve pregnant women. Given the large number of fetuses exposed to DTG or RAL during gestation and ever emerging INSTIs such as bictegravir (BIC) and cabotegravir (CAB), evaluation of safety profile of INSTIs during pregnancy is of importance to ensure limitation of any long-term adverse effects.

Gestation is the critical period of brain development that is likely affected by ARVs [8,20,21,22]. Indeed, prior studies have suggested that DTG, in particular, may be associated with birth defects (NTDs) and other postnatal neurodevelopmental abnormalities [3,11,23,24,25,26]. Moreover, there is limited data available on the safety of BIC and long acting CAB (CAB LA) when used for both treatment or prevention given during pregnancy or at the time of conception. Interestingly, clinical reports of adverse events associated with INSTIs in adults have showed that these ARVs carry class effect. Currently, around 1.3 million HIV-1-infected pregnant women give birth each year, globally [11,12], and the majority of these children will be exposed to INSTIs. Therefore, it is prudent to consider potential mechanisms of INSTI-related neurodevelopmental outcomes. To this end, the current review discusses current United States Food and Drug Administration (U.S. FDA) approved ARVs and recommendations for their usage during pregnancy. In addition, updates are provided on pharmacokinetics (PK) and known INSTI adverse events. 

## 2. Fetal Neurodevelopment

Rapid structural and functional day to day changes occur during early brain development [27]. Insults to the biological, metabolic, or physiological processes during development can have long-term adverse effects (Figure 2). Neurulation begins four weeks after conception with the closure of the neural tube [8,28]. At approximately gestational week 4, neurogenesis begins and involves neural stem cell differentiation into neurons lasting through gestation and beyond to adulthood [8]. Shortly after the initiation of neurogenesis, microglia enter the central nervous system (CNS) where these cells form a resident innate immune population [29,30]. Synaptogenesis begins at 12 weeks with concurrent pruning of synapses and dendritic spines [8]. At 18 weeks, cell populations begin to be refined through apoptosis ensuring proper synaptic connectivity [8,28]. Gliogenesis process begins at gestational week 22 and at week 30, myelination begins; processes that continue into adulthood [8,31].

Brain development is susceptible to drugs, microbes, and inflammatory processes. Disturbances during brain development can result in neural tube defects (NTDs) or postnatal behavioral effects (Figure 2). While it can be difficult to assess cause-effect relationships in observational human gestational studies, the use of animal models in research has proven helpful in evaluating ARV exposures. The sequence of key events in brain maturation is similar between species [21]. Notably, genes crucial for neurodevelopment are conserved between humans and rodents [32]. However, drug dosing based on unique rodent metabolism, timing of administration, and achieving human therapeutic levels is key to the use of rodent pregnancy models for testing ARV safety during pregnancy [21,33].

## 3. Antiretroviral Drugs (ARVs)

The human immunodeficiency virus type one (HIV-1) is a lentivirus belonging to the family Retroviridae, subfamily Orthoretrovirinae [34]. HIV-1 virion contains two copies of the RNA genome and several enzymes, such as reverse transcriptase, integrase, and protease. These enzymes play an important role in making new copies of HIV-1 and are the ARV targets. HIV-1 replication cycle is classified into the following nine major steps include binding, fusion, reverse transcription, integration, transcription, translation, assembly, budding and maturation (Figure 3) [35,36]. Detailed explanation of HIV-1 life cycle is explained by others [35,36].

Each step is a potential drug target and based on these, ARVs are classified into different classes (Figure 3) [37,38]. *First*, entry inhibitors prevent HIV-1 from binding to the target receptors and inhibit entry of the virus into the CD4+ immune cell [37,38]. *Second*, fusion inhibitors prevent the fusion of virus envelope with the CD4+ cell membrane. Drugs can bind to the viral envelope glycoprotein and prevents the conformational changes required for the fusion of viral and cellular membranes. *Third*, nucleoside reverse transcriptase inhibitors (NRTIs) and non-nucleoside reverse transcriptase inhibitors (NNRTIs) block reverse transcriptase function, inhibiting reverse transcription of viral RNA into DNA. NRTIs inhibit reverse transcriptase function by causing chain termination. They compete with natural nucleosides (such as dTTP, dCTP, dGTP and dATP) and block reverse transcription following incorporation into viral DNA. NNRTIs directly and non-competitively bind to reverse transcriptase and inhibit the enzyme’s function by inducing a conformational change [37,38]. *Fourth*, INSTIs block the HIV-1 integrase from incorporating viral DNA into the host cell genome. *Fifth*, protease inhibitors (PIs) inhibit HIV protease, which cleaves new viral proteins and prevent generation of mature viruses. *Sixth*, PK enhancers increase the effectiveness of other ARVs. These act by inhibiting the metabolizing enzyme, cytochrome P450 CYP3A4, that is responsible for metabolizing most PIs [37,38]. Thus, PK enhancers can simplify dosing regimens.

## 4. ARVs during Pregnancy

Recommendations for the use of ARVs for infected pregnant women or for infected women who are planning to conceive, are set by the United States Department of Health and Human Services (DHHS) Panel on Treatment of HIV During Pregnancy and Prevention of Perinatal Transmission, updated on March 17, 2022 [4]. This categorizes individual and combination ARVs into two major categories: *Preferred* and *Alternative*. Herein, we discuss the current recommended treatment guidelines for use of regimens or regimen-backbones according to their respective class in drug-naïve or -experienced women during pregnancy. Recommended categorization of individual ARVs as *preferred, alternative,* or *not recommended* for pregnant women is provided in Table 1.

ARVs or regimens are designated as *Preferred* for treatment during pregnancy, when data in adults have demonstrated efficacy with acceptable adverse events. In addition, PK data during pregnancy are available to decide dosing parameters and the available data must suggest a favorable risk-benefit balance. Recommended *Preferred* dual NRTI backbones include abacavir (ABC) and lamivudine (3TC), tenofovir alafenamide (TAF) and emtricitabine (FTC), or tenofovir disoproxil fumarate (TDF) and FTC. Apart from these available co-formulations, separate doses of TAF and 3TC or separate doses of TDF and 3TC can be administered as an NRTI-backbone once daily. However, weight gain in women on a TAF/FTC backbone should be monitored, especially when used in combination with DTG, as use has been associated with fewer adverse birth outcomes and slightly higher gestational weight gain. Use of TDF should be monitored and avoided in those with renal insufficiency due to the potential renal toxicity. Also, ABC-containing regimens should not be used in infected people who test positive for HLA-B*5701 due to the risk of hypersensitivity reaction associated with ABC. The only recommended *alternative* dual-NRTI backbone is zidovudine (ZDV) and 3TC. Even though this NRTI combination is not recommended for initial therapy in nonpregnant adults, it has been widely utilized in pregnancy. The disadvantages of this backbone include requiring twice daily dosing and the potential risk of hematologic and other toxicities.

From the INSTI class, DTG and RAL are recommended as *preferred* in combination with a preferred dual-NRTI backbone. Both DTG and RAL are *preferred* ARVs for pregnant women who present for initial therapy late in pregnancy because both have shown to rapidly decrease viral load during such settings. DTG use during pregnancy received red flags for a while, due to its potential association with an increase in the risk of NTDs following usage at conception. However, currently, DTG is recommended as a *preferred* ARV throughout pregnancy and for women who are trying to conceive. This is because the most updated data from the Tsepamo study (Botswana) suggested that the prevalence of NTDs with periconceptional usage of DTG was lower than what was reported in interim analysis of the study. The prevalence of NTDs was, still, slightly higher with periconceptional usage of DTG compared with other ART usage at conception, 3 per 1000 deliveries versus 1 per 1000 deliveries, respectively. Based on these data indicating a lower rate of NTDs and favorable risk-benefit balance, the DHHS Panel removed DTG-specific caution and recommended the use of DTG during pregnancy. RAL is preferred during pregnancy, but a twice-daily dosing is recommended. For currently marketed two drug regimens, DTG/3TC and DTG/oral RPV, limited data exists on their usage during pregnancy. However, their components are recommended ARVs for use in pregnancy. Therefore, pregnant women, who present to care on either of these regimens and have successfully maintained viral suppression, can continue the use of it with more frequent viral load monitoring throughout pregnancy. Bictegravir (BIC) in combination with TAF/FTC is approved for use in adults, however, it is not recommended in ART naïve pregnant women due to limited data of BIC in pregnancy. Similarly, an elvitegravir (EVG)/cobicistat (c)-based regimen is not recommended during pregnancy due to limited data and reported reduction in levels of EVG in the second and third trimesters and thus, increased risk of virologic failure during the second and third trimesters of pregnancy. Finally, from the protease inhibitor (PI) class, Atazanavir (ATV)/ritonavir (r) and darunavir (DRV)/ritonavir (r) are recommended as *preferred* ARVs in combination with a *Preferred* dual-NRTI backbone. However, as a precaution, for DRV/r, twice daily dosing is required during pregnancy. Moreover, neonatal bilirubin monitoring is recommended following ATV/r use during pregnancy as ATV is associated with maternal hyperbilirubinemia.

ARVs or regimens are designated as *Alternative* for treatment during pregnancy when data in adults show efficacy and favorable data during pregnancy. However, the data during pregnancy is limited in terms of PK, dosing, formulation, administration, tolerability, or drug-drug interaction in comparison to those of *preferred* ARVs. Currently, no NNRTI is designated as *preferred* for use during pregnancy, in ARV-naïve or ARV-experienced women. Efavirenz (EFV) and RPV are recommended as an *alternative* in combination with a preferred dual-NRTI drug backbone. Even though no evidence has been found of an increased risk of birth defects in fetuses in human studies and extensive available data in human pregnancy, caution is recommended. Moreover, assessment of mother for prenatal and postnatal depression is recommended due to known EFV-associated neuropsychiatric adverse events. For oral RPV-based regimens, frequent viral load monitoring is recommended as PK data during pregnancy suggest lower drug levels and thus, risk of increase in viral load in the second and third trimesters. RPV is also not recommended in situations wherein pretreatment viral RNA copies are more than 100,000 per mL or CD4 cells are less than 200 per mm^3^.

A few ARVs are categorized as *not recommended* for usage during pregnancy. This is due to limited data about efficacy, PK, and safety related to these ARVs during pregnancy. Bictegravir (BIC) in combination with TAF/FTC is approved for use in adults, however, it is *not recommended* in ART naïve pregnant women due to limited data of BIC in pregnancy. Similarly, an elvitegravir (EVG)/cobicistat (c)-based regimen is not recommended during pregnancy due to limited data and reported reduction in levels of EVG in the second and third trimesters and thus, poses increased risk of virologic failure during the second and third trimesters of pregnancy. The newly available combination of RPV and CAB long-acting (LA) is designated as *not recommended* during pregnancy due to insufficient data on women who become pregnant or who were planning to conceive while on this regimen. Thus, pregnant women who are on LA CAB or RPV are recommended to switch to a recommended *preferred* or *alternative* three-drug regimen in pregnancy. In addition, DRV/c and ATV/c are designated as *not recommended* for use during pregnancy due to reductions in drug levels, viral breakthrough and limited data in pregnancy. Lopinavir (LPV)/r is *not recommended* as part of initial regimen in ART naïve women during pregnancy but can be used in special circumstances in treatment experienced women during pregnancy. If LPV/r is used during pregnancy, it should be administered twice daily.

Altogether, ARV regimens used during pregnancy should be carefully chosen based on use history, comorbidities, drug-resistance, and personal preference. Nonetheless, it is suggested that individual or combination ARV benefits and adverse events need be considered. Generally, women given fully suppressive ARVs should continue their use during pregnancy. Discontinuation of a regimen, which is well tolerated, safe, and effective in controlling viral replication, could induce viral rebound, a decline in immunity, and risk fetal HIV-1 transmission. However, a change in ongoing ARV viral suppression may be considered when the treatment regimen includes drug(s) not recommended for use during pregnancy due toxicity and efficacy concerns or not recommended in adults. If infected women achieve virologic suppression and become pregnant while receiving ARVs evaluations need be made on whether to continue or adjust the regimen. If a decision is made to continue with the same ARV regimen, more frequent viral load monitoring should be performed. Finally, for infected pregnant women or who are planning to conceive experiencing virologic failure and have an HIV-1 RNA copy number more than 1000 per mL, resistance testing should be performed to select a new regimen with a greater likelihood of suppressing viral replication to undetectable levels.

## 5. INSTI PK Profiles during Pregnancy

Optimal ARV PK profiling is required to prevent vertical transmission of HIV-1 and improve maternal health [4]. From INSTIs class, RAL and DTG are, currently, classified as *Preferred* ARVs for the treatment of HIV-1 pregnant women. Whereas EVG/c, BIC and CAB are not recommended during pregnancy due to insufficient data about ARV usage in pregnancy. INSTIs are known for their potency in terms of rapid viral suppression and genetic barriers to drug resistance properties in HIV-1 infected adults [14]. Due to their known high efficacy properties in adults, their usage in pregnancy in the future is expected. Nonetheless, knowledge gaps exist on optimal PK profiles of these drugs during pregnancy. In general, physiological changes occurring during pregnancy can drastically reduce drug levels to be below therapeutic requirements. Such change in PK profile can lead to virologic failure. Although levels of INSTIs have been found to be slightly lower during pregnancy, a change in dose is not recommended for DTG or RAL (Table 2). Observed lower drug levels of INSTIs could be due to an increase in progesterone or other hormones during pregnancy, leading to the induction of UGT1A1 or CYP3A4-mediated metabolism [39,40]. Altered drug transporter activity during pregnancy, including p-glycoprotein and breast cancer resistance protein, may lead to increased efflux, contributing to a decrease in maternal drug levels [40]. Although sufficient drug levels are necessary to avoid vertical transmission of HIV-1, direct exposure of the fetus to ARVs in utero may lead to drug-related adverse events and developmental malformations such as neurodevelopmental impairment or structural abnormalities [3,11,23,24,25,26,41,42]. Thus, in addition to PK of each INSTI in the mother during pregnancy, we discuss transplacental drug levels of each INSTI and its subsequent elimination half-life in neonates (Table 2).

### 5.1. Raltegravir (RAL)

RAL was the first drug approved from the INSTI class [14]. It is mainly metabolized by UDP glucuronosyltransferase family 1 member A1 (UGT1A1) [54]. PK parameters of RAL during pregnancy at 400 mg twice daily was described [43]. The PK profile of RAL is shown to have extensive inter-subject variability during pregnancy. Such inter-or-intra-subject variability in PK parameters of RAL was also observed in non-pregnant adults [43,44,55], suggesting such PK variability is not influenced by pregnancy.

Delays in absorption of RAL in 38, 32 and 12% of women were reported in the 2nd trimester, 3rd trimester and postpartum, respectively [43]. Median maximum concentration (C_max_) values of RAL were slightly lower in the 2nd (0.365–5.960 µg/mL) and 3rd trimesters (0.315–7.820 µg/mL) compared to postpartum (0.312–12.600 µg/mL). This showed 26 and 41% lower C_max_ in the 2nd and 3rd trimester respectively. Moreover, median concentration 12 h after last dose (C_12_) was significantly lower in the 2nd trimester (0.0128–0.438 µg/mL) in comparison to postpartum (0.0199–1.340 µg/mL). Further, RAL exposure (AUC) was reduced by approximately 50% during the 2nd and 3rd trimesters of pregnancy than during postpartum. Median AUC_0–12_ was 6.6 µg*h/mL in the 2nd trimester and 5.4 µg*h/mL in the 3rd trimester compared to 11.6 µg*h/mLduring the postpartum period. Similar to these observations, RAL exposure (AUC), C_max_ and C_12_ were found to be reduced by 29, 18 and 36%, respectively during the 3rd trimester of pregnancy in comparison to postpartum [44]. Induction of the UGT1A1 pathway and increased expression of liver P-glycoprotein (P-gp) during pregnancy may have contributed to decreased exposure to RAL during pregnancy compared to postpartum [40]. Neither study saw differences in half-life (T_1/2_) of RAL during pregnancy and postpartum. Recently, total and unbound levels of RAL levels were recorded during pregnancy [45]. RAL shows moderate protein binding at approximately 76 to 83% [56]. This study showed that exposure (AUC) of median total and unbound fraction of RAL (active form of RAL) during the 3rd trimester was significantly decreased by 36 and 27%, respectively, in comparison to postpartum levels [45]. Median total trough concentrations (C_trough_) of RAL were significantly lower (28%) during pregnancy. However, unbound C_trough_ levels of RAL showed a non-significant decrease during pregnancy compared to postpartum, signifying less of an effect of pregnancy on unbound levels of RAL compared to total RAL levels [45].

RAL was found to readily cross the placenta and median ratios of RAL cord blood to maternal blood were 1.5 and 1.21, respectively [43,44]. In addition to these studies, other clinical studies also reported high placental transfer of RAL with overall cord blood to maternal blood ratios varying from 1 to 3.48 [43]. Consistent with clinical data, placental transfer of RAL was found to be high in an open-circuit ex-vivo human cotyledon perfusion model with a fetal transfer rate (FTR) of 9.1% ± 1.4% (mean ± standard deviation) [57]. Also, concentrations of RAL were measured in infants following transplacental transfer from HIV-1 infected mothers. RAL levels in infants within three hours after delivery were seven to nine times higher than maternal paired drug levels [58]. These data were in line with another case report analyzing PK of RAL in a premature newborn [59]. This study reported 29 ng/mL of RAL in plasma of the infant one month after transplacental exposure, and the calculated half-life of drug in this premature infant was approximately 168 h. Furthermore, RAL elimination in infants was found to be highly variable and prolonged. A median elimination T_1/2_ was 26.6 h (9.3–184 h) in neonates (N = 17) following transplacental transfer [53]. High concentration and prolonged elimination of RAL from infants following transplacental transfer was likely due to immaturity of UGT1A1 enzyme activity in infants [53,58].

### 5.2. Elvitegravir (EVG)

EVG is a first generation INSTI [14]. Unlike other INSTIs, EVG is primarily metabolized by CYP3A4 and with minor contribution from UGT1A1 pathway [60]. In clinical practice, elvitegravir is co-administered with the PK enhancer cobicistat (c), a potent mechanism-based inhibitor of CYP3A4, in order to increase EVG exposure and facilitate once-daily dosing. EVG/c is available in 150 mg/150 mg dosage for once daily oral administration.

EVG exposure (AUC_0–24_) was shown to be significantly reduced in the 2nd trimester (24%) and in the 3rd trimester (44%) in comparison to paired exposure levels during the postpartum period [46]. Median EVG AUC_0–24_ levels were 15.283 (11.939–19.038) µg*h/mL, 14.004 (9.119–18.798) µg*h/mL and 21.039 (13.532–32.788) µg*h/mL, in the 2nd trimester, 3rd trimester and postpartum period, respectively. Median EVG C_max_ was not significantly different in the 2nd trimester, however, it was 28% lower in the 3rd trimester (1.432 µg/mL) compared to postpartum (1.713 µg/mL). EVG C_24_ concentrations were 81% and 89% lower in the 2nd and 3rd trimester, respectively, compared to postpartum levels. Median EVG C_max_ concentrations were 0.025 µg/mL, 0.048 µg/mL and 0.377 µg/mL in the 2nd trimester, 3rd trimester and postpartum. In agreement with these data, others reported 34% reduced EVG exposure and 77% lower C_trough_ levels in the 3rd trimester of pregnancy compared to paired values at postpartum [47]. The geometric mean of EVG C_max_ was lower in the 3rd trimester (1.4 µg/mL) compared to that of postpartum (1.8 µg/mL). Moreover, both studies reported a significant decrease in half-life (T_1/2_) of EVG during pregnancy compared to the postpartum period. In a case report, total and unbound levels of EVG in an HIV-1-infected woman during the 3rd trimester of pregnancy and postpartum was recorded [48]. Elvitegravir is highly protein-bound in plasma (98–99%), primarily to albumin [60]. (AUC_0–24_), C_min_ and half-life (T_1/2_) for total EVG were considerably lower during the 3rd trimester of pregnancy than reference values. Consequently, this study showed that C_min_ and C_0h_ of unbound EVG was less than 0.0005 µg/mL, and therefore suboptimal levels of unbound EVG (active form) during pregnancy.

High placental transfer of EVG during pregnancy was reported with the ratio of cord blood to maternal blood at 0.91, 0.75 and 0.64, respectively [46,47,48]. In a case report, EVG was found to readily pass through the placenta during pregnancy with a ratio of 1 for cord blood to maternal blood EVG concentrations [61]. In addition to in vivo data, a study utilizing an ex-vivo human dually perfused cotyledon model with open and closed circuit reported moderate placental transfer of EVG with FTR of 19% ± 13% and 20% ± 10% (mean ± SD), respectively [62]. Furthermore, the median elimination half-life of EVG in infants was found to be 7.6 h (n = 23) following in utero transfer.

### 5.3. Dolutegravir (DTG)

DTG is a second generation INSTI [14]. It is mainly metabolized by UGT1A1 and secondarily by CYP3A4 [63,64]. Detailed PK profile of 50 mg once daily DTG during pregnancy was reported in [49,50,51]. Median DTG AUC_0–24_ was 37 and 29% lower in the 2nd and 3rd trimesters in comparison with postpartum [49]. Median drug AUC_0–24_ was 47.6 µg*h/mL in the 2nd trimester, 49.2 µg*h/mL in the 3rd trimester and 65 µg*h/mL in the postpartum period. DTG C_max_ was 26% and 25% lower in the 2nd (2.57–4.63 µg/mL) and 3^rd^ trimesters (2.66–4.24 µg/mL), respectively, compared to paired postpartum concentrations (3.83–5.97 µg/mL). DTG C_24_ levels were 0.63–1.34 µg/mL, 0.68–1.34 µg/mL and 0.80–1.95 µg/mL in the 2nd and 3rd trimester and postpartum, respectively. Thus, DTG C_24_ levels were 51% and 34% lower in the 2nd and 3rd trimester compared to paired postpartum levels. Similarly, others noted reduction in DTG exposure (AUC), C_max,_ and C_24_ or C_trough_ in the 3rd trimester of pregnancy compared to postpartum on once-daily dosing [50,51]. Such a decrease in DTG exposure is likely due to the induction of UGT1A1 and CYP3A4 pathways and increased expression of liver P-gp during pregnancy [40]. Studies also observed a significant decrease in DTG half-life (T_1/2_) compared to postpartum [49,51]. In addition to total DTG quantitation, unbound DTG levels were recorded during pregnancy [51]. DTG showed approximately 99% binding to plasma proteins, mainly to albumin [40,65]. This study reported comparable levels of unbound drug between pregnancy and postpartum, suggesting no impact of pregnancy on unbound DTG concentrations. This is of importance as unbound DTG is the pharmacologically active form. C_max_ of unbound DTG was 12.3 and 9.18 µg/L and C_min_ of unbound DTG was 2.87 and 3.38 µg/L during the 3rd trimester and postpartum period, respectively.

Studies found high placental transfer of DTG and median ratios of drug levels in cord blood to maternal blood were 1.25, 1.21, and 1.29 [49,50,51]. Moreover, two different studies utilizing a closed circuit ex- vivo human dually perfused cotyledon model confirmed high DTG transfer from mother to fetus with a mean fetal-to-maternal ratio of 34 and 60% [42,66]. In affirmation, drug accumulation was confirmed with prolonged DTG elimination from infants [49,50]. Furthermore, the median elimination half-life of DTG was 32.8 h in infants (n = 16) following transplacental transfer in utero [49]. Whereas a median postpartum elimination half-life in paired mothers was 13.5 h. These PK data were consistent with a case report of PK of DTG in a premature neonate [67]. This study described a 4-fold increase in elimination half-life of DTG in the infant compared to adults [67]. The calculated terminal half-life of DTG in this infant was approximately 46 h, based on a one-phase PK elimination model. Like RAL, prolonged elimination of DTG from infants following transplacental transfer may be due to low activity of UGT1A1 enzyme activity at birth [49].

Our own work determined PK and biodistribution (BD) profiles of DTG during pregnancy in mice following daily oral administration of supratherapeutic dosage that can achieve 10× the C_max_ recorded in humans [23]. High DTG levels were observed in maternal plasma (average 27,296.3 ± 2701.15 ng/mL) and paired placenta (average 3528.04 ± 316.56 ng/g) at gestation day (GD) 16.5. The ratio of placental DTG levels to mother’s plasma DTG levels was 0.12. Interestingly, high DTG concentrations in embryonic brain tissue (663.6 ± 75.13 ng/g) were measured at GD 16.5. DTG levels were also detectable in brain tissue of postnatal pups at PND 4 (48.1 ± 2.35 ng/g at PND 4) after cessation of maternal DTG administration at the day of birth of pups. At GD 16.5, the ratio of embryo brain DTG levels to mother’s plasma DTG levels was 0.024 and ratio of embryo brain DTG levels to placenta DTG levels was 0.18. Overall, our study confirmed and was consistent with previous clinical reports of high placental transfer of DTG to fetuses, drug accumulation and prolonged elimination from neonates. Most importantly, it was the first study to show DTG concentrations in the developing embryo brain during pregnancy, identifying direct exposure of the embryo brain to DTG during a critical period of development. This is of utmost importance due to the association of DTG with NTDs and postnatal neurologic abnormalities.

### 5.4. Bictegravir (BIC)

BIC is a second generation [14] and one of the newest INSTIs to be approved to treat HIV-1 infection [52,68]. BIC is mainly metabolized by CYP3A4 and UGT1A1 [52,68]. Currently, this drug is not recommended for use during pregnancy due to insufficient data on PK, efficacy, and safety during pregnancy. The following two case studies report BIC PK in pregnancy. In the first case study, BIC PK results were reported during pregnancy and the postpartum period [52]. In an infected 33-year-old pregnant women, during the 3rd trimester, BIC AUC_0–24h_, C_trough_ and C_max_ were 35%, 49% and 19% lower, respectively, compared to PK data 6 weeks following postpartum. During pregnancy, BIC AUC_0–24h_, C_trough_, and C_max_ concentrations were 37.9 µg*h/mL, 0.63 µg/mL and 3.82 µg/mL, respectively. Whereas, during the postpartum period, BIC AUC_0–24h_, C_trough_ and C_max_ concentrations were 58.0 µg*h/mL, 1.23 µg/mL and 4.7 µg/mL, respectively. The observed modest decrease in PK of BIC during the 3rd trimester of pregnancy compared to postpartum may be due to induction of the UGT1A1 and CYP3A4 pathways during pregnancy. Further, same study reported high transplacental transfer of BIC during pregnancy. The ratio of cord blood to maternal blood was 1.49. Observed high transfer of BIC through placenta contrasted with reported low transfer of BIC in an open-circuit ex-vivo human dually perfused cotyledon model with median maternal-to-fetal ratio of 7% (6–9.5%) [69].

Recently, total and unbound levels of BIC were examined in plasma of a 35-year-old woman, coinfected with HIV-1 and HBV [70]. After initiating the BIC-based regimen at 29 weeks of gestation, at delivery, maternal plasma total and unbound BIC concentrations were 2.826 and 0.017 µg/mL. Cord blood total and unbound BIC plasma concentrations were 1.933 and 0.010 µg/mL, respectively. These values of total BIC corresponded with the ratio of cord blood to maternal blood at 0.68. This study showed that plasma levels of total BIC were above protein adjusted EC95 (0.162 µg/mL) throughout the 3rd trimester of pregnancy (starting from the 29th week of gestation) and at the time of delivery. However, this study did not report postpartum concentration in the patient, so intra-patient comparison was not available. In addition to maternal BIC levels during pregnancy, plasma BIC levels were evaluated in the neonate following gestational transfer. On day 3 after birth, plasma BIC concentration in the neonate was 2.097 µg/mL and went under the limit of detection (LOQ; 5 ng/mL) on day 22. Three days after birth, BIC concentrations in plasma of the neonate remained similar to plasma concentrations at birth, suggesting slow elimination of drug. Such slow elimination of BIC from the neonate following transplacental transfer could be due to the immaturity of CYP3A4 and UGT1A1 at birth [70].

### 5.5. Cabotegravir (CAB)

CAB is the newest INSTIs to be approved to treat and prevent HIV-1 infection [71,72]. It is available in an oral tablet for daily administration and long acting injectable nanosuspension for monthly or bimonthly administration [71,72]. CAB is primarily metabolized by UGT1A1 with a minor contribution from UGT1A9. Similar to other INSTIs, it shows high binding to plasma proteins (>99%) [73]. Currently, no data is available about CAB usage in human pregnancy. Therefore, CAB is not recommended for usage during pregnancy. However, placental transfer rate of CAB in an open-circuit ex-vivo human dually perfused cotyledon model was reported [69]. Low CAB transfer across the placenta with a median maternal-to-fetal ratio of 10% (5–16%) was observed and discussed potential limitations of CAB as pre-exposure prophylaxis in fetuses due to low transfer [69]. The same study also reported low transfer of BIC, which was in contrast with high transplacental transfer of BIC in a human case report [52,69]. Therefore, more investigation and comprehensive data are required to understand the PK, placental passage, and safety of CAB in pregnancy.

## 6. INSTI Adverse Events

### 6.1. Neuropsychiatric Adverse Events

Early in the study of HIV-1, it was evident that HIV-1 had the ability to enter the CNS and lead to HIV-associated neurocognitive disorders (HAND). The use of ART has significantly improved patient outcomes and reduced the HAND severity. However, mild cognitive impairment has persisted in some patients, suggesting potential CNS toxicity of ARVs. One of the adverse events (AE) associated with such toxicity are considered neuropsychiatric adverse events (NPAEs). Several clinical studies have reported that INSTI use in adults is linked with NPEAs, ranging from insomnia to severe depression [74,75,76]. These INSTIs-linked NPAEs have been found to have higher rates than initially suggested by clinical trials. INSTIs have also been associated with impaired learning and memory and lower brain volumes [77,78,79].

Of approved INSTIs for clinical usage, DTG has been associated with a higher incidence of NPAEs and subsequent discontinuation in comparison to other drugs of this class [80,81,82,83,84,85]. In a retrospective analysis of patients initiating an INSTI at two German clinics, the rate of NPAEs leading to discontinuation within 12 months was 5.6% for DTG, 1.9% for RAL, and 0.7% for EVG [81]. Of the patients who discontinued DTG due to NPAEs, the median time between initiation and discontinuation was 3.1 months and 78% of these patients stopped DTG use within 6 months. This and other studies confirm that discontinuation of DTG resulted in significant improvement of these symptoms [74,81,84]. Common neuropsychiatric symptoms reported with DTG use include insomnia, abnormal dreams, and depression [86]. Scheper et al. reported that when DTG initiation-related mild depression was not addressed in a male patient, his symptoms worsened over time. Four months after initiation of a DTG-based regimen, he developed serious suicidal thoughts, which were relieved with a switch to a different regimen [74]. Moreover, Kanai et al. found that continuation of DTG use after the emergence of minor NPAEs can lead to schizophrenic brief psychotic disorder, and persistent cenesthopathy, which was not found to be relieved by DTG cessation [87]. These studies highlight the importance of vigilant monitoring for NPAEs, especially with ARV regimens that include DTG.

Although reports of NPAEs associated with INSTIs suggest neurotoxic effects, underlying mechanisms are not yet uncovered. Multiple risk factors for DTG-related NPAEs have been identified. Women, patients older than 60 years, patients initiating abacavir (ABC) at the same time as DTG, and patients who are ART-experienced may be more vulnerable to experiencing NPAEs [76,81,82,88,89,90,91]. Studies have associated variations in the UGT1A1 allele, including UGT1A1*6 and UGT1A1*28, both with higher trough DTG concentrations and a higher cumulative incidence of neuropsychiatric adverse events [87,92]. This association between DTG concentrations and NPAEs may be key to understanding why some populations are more susceptible to NPAEs than others. Updated PK data that includes underrepresented populations may elucidate the mechanisms behind DTG neurotoxicity. Moreover, a clinical study reported high DTG concentrations in the cerebrospinal fluid (CSF) of HIV-1-infected, ARV-naive patients, suggesting a potential mechanism through drug-linked neurotoxicity events [93].

Due to similarities in structure and mechanism of action, CNS toxicity leading to NPAEs may be a class effect [86,94]. While DTG has been highlighted for its association with NPAEs, other INSTIs have been investigated. RAL has been associated with the exacerbation of pre-existing depression, and concentration-related insomnia and nightmares [95,96]. In a prospective cohort study that included only female patients, use of elvitegravir was associated with poorer learning, suggesting cognitive impairment [78]. For BIC, clinical trials found that rates of NPAEs were comparable to DTG [97,98]. However, more data are needed for BIC and CAB to determine if their usage is associated with NPAEs. Overall, current data suggests that INSTIs, and particularly DTG, are associated with NPAEs and physicians should carefully monitor patients with or without history of neuropsychiatric symptoms treated with these regimens.

### 6.2. Weight Gain

Due to the widespread inflammation associated with persistent viremia, HIV infection is frequently associated with wasting syndrome. With ART initiation, viremia and the associated inflammation subside, leading to weight gain. In normal or underweight patients, this is referred to as a “return to health”. However, with many patients initiating ART in the overweight or obese categories, this weight gain could also bring a number of weight-related illnesses. INSTIs in particular are of interest due to their association with weight gain and obesity over other classes of ARVs [99,100,101]. Compared to women initiating treatment with PIs, a higher proportion of women who initiated an INSTI treatment had >5% weight gain with a mean difference in weight gain of 1.5 kg [101]. These women were also more likely to shift to a worse BMI classification than before initiating INSTI use. Women enrolled in the Women’s Interagency HIV Study switching to or adding an INSTI to their treatment had a greater mean increase in body weight of 2.1 kg compared to women on a non-INSTI regimen [102]. In another study, patients on an INSTI treatment gained more weight than those on an NNRTI or PI regimen, gaining an average of 3.24 kg with INSTIs, 1.93 kg with NNRTIs, or 1.72 kg with PIs [100]. This study also suggested that BIC and DTG were associated with more weight gain than cobicistat-boosted EVG (BIC, +4.24 kg; DTG, +4.07 kg; EVG/c, +2.72 kg). Similarly, Bourgi et al. found that people with HIV initiating an INSTI regimen had a mean estimated weight gain over 5 years of 5.9 kg compared to a gain of 3.7 kg for NNRTI and 5.5 kg for PI regimens [103]. In this study, DTG was found to be associated with higher weight gain than the other INSTIs, RAL and EVG (DTG, +7.2 kg; RAL, +5.8 kg; EVG, +4.1 kg). DTG can produce a greater median increase in body weight and lead to more obesity than other ARVs [99,104,105]. This has been found both in adults and children switching to DTG [106]. In a switch study, patients switching to a tenofovir/lamivudine/dolutegravir regimen had a mean weight gain of 3.88 +/− 2.021 kg over one year compared to patients remaining on a TFV/3TC/EFV regimen (2.26 +/− 2.39 kg) [107]. Similarly, others found that patients who switched to a TLD regimen went from gaining 0.35 kg/year to 1.46 kg/year in the year after transitioning [108]. These studies suggest that higher weight gain is associated with INSTI regimens and some INSTIs, like DTG, may have a greater effect. Moreover, weight gain should be universally reported in association with INSTI use. Weight gain may be under-reported in underdeveloped areas due to positive perceptions of weight gain, leading to inefficient risk communication to HIV patients experiencing unhealthy weight gain [109].

Baseline risk factors for excess weight gain with any ART regimen include female sex, race, low CD4+ cell counts and high HIV RNA levels [100]. Women seem to be at a higher predisposition to unwanted weight gain following INSTI treatment [104,107,110]. Chen et al. found that African American women starting an INSTI regimen were more likely to shift from normal or overweight BMI classification to a worse classification compared to a PI regimen [101]. In treatment naïve patients, a baseline CD4 count of <200/µL was associated with an average weight gain of 2.97 kg more than patients with a baseline CD4 count >200/µL. In the same study, a baseline HIV RNA level of >100,000 copies/mL was associated with 0.96 kg higher weight gain compared to patients with lower HIV RNA levels [100].

One of the most prominent risk factors for weight gain associated with INSTI use is combination with TAF. It is unknown whether this effect is due to TAF alone or its interactions with drugs in the INSTI class. In a trial in South Africa, patients who had not had any ART within the past 6 months and never been treated with any form of ART for more than 30 days were assigned to a regimen including TAF/FTC/DTG (TAF-based), TDF/FTC/DTG (TDF-based), or TDF-FTC-EFV (standard-care). Patients in the TAF-based group gained an average of 6.4 kg while the TDF-based and standard-care groups gained an average of 3.2 kg and 1.7 kg, respectively [104]. Another study found that people with HIV switching from TDF to TAF and maintaining a regimen that included DTG had a sudden increase in weight. For these patients, the average rate of weight gain on TDF was 0.73 kg/year, which increased to 2.38 kg/year in the 9 months following a switch to TAF [111]. In this study, maintaining EVG/c and RAL were also associated with an increase in weight gain. These studies suggest that use of TAF in combination with an INSTI can lead to significant weight gain in both treatment-naïve and experienced individuals. ARV drug regimens should be carefully curated to avoid such drastic changes in weight gain.

### 6.3. Diabetes Mellitus

Emerging evidence indicates that INSTI-based ART treatment may be associated with metabolic outcomes such as diabetes mellitus in HIV-1 infected people. Early evidence for INSTIs association with diabetes was conflicting. The Women’s Interagency HIV study, a cohort study, did not observe significant increases in the incidence of diabetes mellitus associated with INSTI-based regimens. However, in the same cohort, a significant increase in hemoglobin A1c (HbA1c) was found in patients treated with INSTI-based regimens versus those with non-INSTI-based regimens, suggesting a possible impact of INSTIs on glycemic control [112]. Notably, HbA1c increases were among women who also experienced clinically significant (≥5%) body weight gain [112]. Also, a large French cohort study did not find statistically significant association of INSTIs with new-onset diabetes mellitus in comparison to NNRTI or PIs [113].

Now, studies suggest INSTIs affect glucose metabolism. The risk of new-onset diabetes mellitus/hyperglycemia in HIV-1 infected patients was evaluated during the first six months following the initiation of INSTI-based regimens [114]. A comparative study was performed to evaluate the effect of INSTI- versus non-INSTI-based ART regimens on the risk of diabetes/hyperglycemia among non-elderly infected patients. Overall, a higher risk of new-onset diabetes/hyperglycemia was reported. HIV-1 infected patients treated with INSTIs were 31% more likely to have new-onset of diabetes mellitus/hyperglycemia compared with those treated with non–INSTI-based regimens [114]. Moreover, evaluation for association with individual INSTIs showed that EVG was associated with the greatest risk of new-onset diabetes/hyperglycemia, followed by DTG and then RAL. Moreover, Nolan NS et al. recently reported 3 cases in which HIV-infected patients developed hyperglycemia and ketoacidosis within months of being switched to a BIC-based regimen [115]. HbA1c values before starting a BIC-based regimen were 5.6–6.6%, however, at the time of presentation with hyperglycemia and ketoacidosis, HbA1c values ranged from 12.4% to 17%. The short time period to the development of diabetic ketoacidosis (DKA) and rapid worsening of HbA1c strongly implicated BIC as an associated factor. In addition, discontinuation of BIC-based ART resulted in significant decreases in insulin requirements [115].

Severe hyperglycemia with or without its life-threatening acute complications (diabetic ketoacidosis or hyperosmolar hyperglycemia state) was recognized after INSTI use in SAILING, SPRING-2, SINGLE, and VIKING-3 clinical trials [116,117,118,119]. Pivotal DTG clinical trials also suggest that extended duration of DTG-based treatment increases the risk for hyperglycemia. SPRING-2 and SINGLE studies reported hyperglycemia (grade 2 and grade 3 combined) in fewer than 7% of treatment naïve participants at 96 weeks and in 11% at 144 weeks. The VIKING-3 study had experienced patients and grade 2–4 hyperglycemia was reported in 14% of patients at week 48. In addition to clinical trials, several case reports have described new-onset or acute worsening of well-controlled diabetes in infected patients after switching to INSTI-based regimens [115,120,121,122,123,124].

The underlying mechanism for the association of INSTIs with diabetes mellitus is still unknown. However, the proposed mechanism has been the medication’s ability to chelate Mg^2+^ at the cellular level, which can affect glucose transport via glucose transporter type 4 (GLUT-4) receptors, causing insulin resistance [121,122]. Moreover, there is now clear evidence that INSTIs can cause weight gain in infected patients, especially in women, which can also contribute to insulin resistance and development of diabetes mellitus [125]. Considering the association of INSTIs with weight gain, there is a risk of developing diabetes. Therefore, clinicians must seek careful consideration while initiating INSTI-based regimen therapy, especially in patients who are overweight at baseline. In conclusion, hyperglycemia is a potential adverse event associated with INSTI-based regimens. Thus, baseline and periodic monitoring of glucose levels might be considered after initiation of INSTI ART regimens.

## 7. Known Side Effects of INSTIs during Development

### 7.1. Neural Tube Defects (NTDs)

In May 2018, concerns emerged for DTG usage in pregnant women or those of child-bearing age. An unplanned interim analysis from the Botswana birth-outcomes surveillance study reported a higher prevalence of NTDs among infants born to women taking DTG-based regimens at the time of conception in comparison to infants born to women taking other ART regimens [3]. Among the 426 infants born to HIV-1-positive women who had been taking DTG-based regimen at the time of conception, 4 (0.94%) infants had NTDs. These 4 NTDs were identified as encephalocele, myelomeningocele (along with undescended testes), and iniencephaly (along with major limb defect). Later in August 2019, additional data from follow up of additional pregnancies from Botswana (Tsepamo study) was reported. Among 1683 infants born to women taking a DTG-based regimen at conception, 5 NTDs were found (0.30%) in comparison to 15 NTDs among 14,792 infants (0.10%) born to women taking any other ARV-based regimen at conception [26]. The absolute difference in prevalence of NTDs between DTG-based and non-DTG-based ART exposure at conception was 0.20%. The identified total 5 NTDs associated with DTG usage at conception included 2 of myelomeningocele, 1 of encephalocele, and 1 of iniencephaly (all diagnosed with photographs), as well as 1 of anencephaly. Moreover, this study reported that in all 5 cases of NTDs, all five women started DTG more than 3 months before the estimated date of conception. Overall, this observational study from Botswana concluded that the prevalence of NTDs was slightly higher in association with DTG exposure at conception than with other types of ART exposure at conception (3 per 1000 deliveries vs. 1 per 1000 deliveries). Apart from the Tsepamo study from Botswana, few other studies reported concerns associated with DTG usage during pregnancy. Other surveillance data from non-Tsepamo health facilities of Botswana showed increased NTD prevalence in women with DTG exposure at conception. One neural-tube defect was found among 152 deliveries (0.66%) [126]. A retrospective cohort analysis of pregnant women with HIV-1 on DTG from two urban clinics in the United States reported a relatively high proportion of preterm births (31.6%) among total 66 deliveries [127]. Later, two cases of NTDs were reported that occurred in women taking a DTG-based ARV regimen at the time of conception in the city of Porto Alegre, Brazil [128]. These two NTDs in male fetuses were identified as anencephaly and spina bifida. Yet, risk of NTDs following usage in early pregnancy is very low and no direct link to DTG has been identified in clinical studies. After the safety signal of DTG associated potential risk of NTDs, a study reviewed two large, independent antiretroviral pregnancy registries to evaluate pregnancy outcomes following maternal DTG treatment during pregnancy. No NTDs were observed in both registries after first trimester DTG exposure [129]. A Canadian surveillance study examined the rates of congenital anomalies in infants born to women living with HIV (WLWH) and their potential associations with DTG using data from the Canadian Perinatal HIV Surveillance Program (CPHSP). This study found no NTDs in the small number (N = 69) of women on DTG at conception [130]. A retrospective, observational, national, cohort study from Brazil, also did not observe occurrences of NTDs in infants born to women (N = 382) during periconceptional DTG exposure [131].

Like clinical observations, animal and cell culture studies reported inconsistent findings. An in vitro rat whole embryo culture study investigated the effect of DTG exposure during the critical period for neural tube development. Rat whole embryo culture was treated with DTG at the maximum human recommended dose. However, no developmental abnormal effects were observed [132]. Moreover, studies performed as part of the medicine development program for DTG included a series of reproductive and developmental toxicity studies. No malformations or other developmental abnormalities were observed in rats or rabbits following exposure to DTG during the period of organogenesis [133]. A study conducted in zebrafish showed developmental toxicity after early DTG embryonic exposures. This study also showed that DTG is a partial antagonist of the folate receptor (FOLR1), and supplemental folic acid can potentially mitigate DTG-associated developmental toxicity in zebrafish [24]. In another study, Mohan H et al. showed that DTG at therapeutic doses is associated with an increased risk for fetal defects, including NTDs in mice [25]. The observed rate of fetal defects in mice was similar to that reported in the Tsepamo study from Botswana. Interestingly, NTDs were not observed at a supratherapeutic dose of DTG [25]. Our own work, even with a small sample size of pregnant mice (N = 17), found a 0.9% incidence of NTDs (1 out of 103 embryos) in embryos exposed to a supratherapeutic dosage of DTG from the time of conception [23]. Recently, developmental toxicity associated with DTG was investigated using a stem cell-based in vitro morphogenesis model. This study observed that DTG impairs morphological and molecular aspects of the in vitro morphogenesis models in a concentration-dependent manner. Also, adverse effects were observed when the morphogenesis in vitro model was exposed to DTG at early stages of development, but not at later stages [134].

To this date, no NTDs were reported linked to any other clinically approved INSTI. Even though there is no consensus among clinical and basic science research in relation to identifying direct association of DTG to NTDs, one observation was consistent in all of them. It was developmental toxicity associated with DTG use at conception or early during pregnancy. Thus, more studies, including those in human, animals, and cell culture models, are needed to evaluate the pregnancy outcomes following periconceptional usage of DTG to establish a cause-and-effect relationship and identify a novel underlying mechanism.

### 7.2. Postnatal Neurodevelopmental Deficits

During the last 3–4 years, significant efforts from clinical and basic science research groups were engaged to study DTG-associated structural defects of the brain and spinal cord and potential underlying mechanisms after the report of an increased risk of NTDs in infants born to mothers taking DTG periconceptionally. Yet there is a research gap of adverse events reflecting DTG-associated postnatal neurodevelopmental impairments. This is, in particular, for babies born without structural malformations. It remains a major knowledge gap, especially when the risk of NTDs is very low (0.2–0.3%) and the majority of infants are born without structural malformations. Indeed, it is now well accepted that DTG causes neuropsychiatric adverse events in adults ranging from insomnia to severe depression [86,135]. DTG-associated neuronal toxicity is also confirmed by in vitro and research animal studies [136,137]. However, as usage of DTG is relatively new during pregnancy and postnatal neurodevelopment requires long-term longitudinal assessments, there is limited data about the effects of DTG on postnatal neurodevelopmental outcomes following in utero drug exposure. Recently, the Surveillance Monitoring for ART Toxicities (SMARTT) study reported an increased risk of postnatal neurological abnormalities during development such as febrile seizures, microcephaly, epilepsy and other neurological disorders following gestational ARV exposures [11]. Longitudinal analysis of the study revealed a strong association of DTG with neurological disorders following drug usage during the first trimester (aRR = 2.95, 95% CI:0.79, 11.01) and at conception (aRR = 3.47, 95% CI:0.74, 16.36). Moreover, the magnitude of association between in utero DTG exposure and neurological disorder status was statistically significant for first trimester exposure of DTG after accounting for years of follow-up (*p* = 0.036) and after accounting for clustering within research site (*p* = 0.048). This clinical study not only underscored the association of in utero DTG exposure with the risk of postnatal neurodevelopmental impairments, but also confirmed other clinical/researchers’ findings that exposure to DTG at conception would have more adverse impact on structural and functional brain development [3,23,25,26]. Moreover, clinical studies have showed high levels of transplacental DTG transferred from mother to fetus [49,50,51]. Moreover, drug accumulation was confirmed with prolonged elimination of DTG from infants following transplacental transfer [49,67] (Table 2). Thus, identifying unknown effects of in utero DTG exposure on postnatal brain development and its extent during childhood development is essential. To this date, there are no reports on effects of other INSTIs on postnatal neurodevelopment following gestational exposure. However, high levels of transplacental transfer of other integrase inhibitors (BIC, RAL or EVG) have been well documented (Table 2). Given the large number of fetuses being exposed to DTG in utero [4,5], it is vital to assure their safety and is timely to identify unknown adverse effects of in utero DTG on postnatal neurodevelopment and determine underlying mechanisms so that intervention strategies can be developed.

## 8. Potential Mechanisms Affecting Neurodevelopment

### 8.1. Maternal Mental Health

As described previously in this review, neuropsychiatric adverse events are a potential risk related to the use of INSTIs to treat HIV-1. These adverse events range from mild insomnia to major depression and suicidal thoughts. Data suggests that women are particularly susceptible to NPAEs. In a German retrospective analysis, Hoffman et al. found that women had a significantly increased risk of DTG discontinuation due to NPAEs, 10.0% of women discontinuing in comparison to only 4.6% of men discontinuing due to NPAEs [81]. It is possible that PK differences of DTG and other INSTIs between women and men may influence DTG concentrations, and therefore NPAEs. With such observations in adult women, it is possible that pregnant or postpartum women could be more susceptible to NPAEs [76].

Although developmental outcomes have not been well studied, NPAEs could have negative effects on the developing fetus. Maternal insomnia is associated with preterm delivery and postpartum depression, which could both have significant effects on the development of the child [138]. Mothers serve as external regulators of neurodevelopment; therefore, postpartum depression can have negative impacts on neurodevelopmental outcomes and contribute to poor cognitive and socio-emotional development [139,140]. Further, maternal distress during the perinatal period can also have negative effects on cognitive, as well as behavioral and psychomotor development [140]. These effects can also be evidenced by structural changes in offspring associated with maternal major depressive disorder [141]. Overall severity of both perinatal and postnatal depression can also directly link to severity of neurodevelopmental disadvantages in the offspring [142]. Given the potential detrimental effects of maternal NPAEs on fetal neurodevelopment and the apparent increased risk of women developing INSTI-related NPAEs, pregnant and postpartum women on an INSTI regimen should be carefully monitored for neuropsychiatric symptoms.

### 8.2. Maternal Weight Gain

Weight loss and wasting are hallmarks of untreated HIV-1 infection, which presents a significant limitation for women planning to become pregnant. Insufficient gestational weight gain has adverse effects on fetal neurodevelopment [143]. However, maternal obesity also has been associated with cognitive problems, ADHD, eating disorders, and psychotic disorders in offspring [144]. This suggests that weight change in either direction could affect fetal neurodevelopment. The use of ARVs has limited weight loss and wasting. However, INSTIs have been associated with excess weight gain with women being more susceptible [104,107,110]. In the Botswana study, weight gain during pregnancy was compared between women initiating EFV and DTG [145]. The mean weekly weight gain between 18 and 36 weeks of gestation for women initiating DTG was 0.05 kg/week higher than that of women initiating EFV. Average weight gain over this period of time was 1.05 kg higher for women initiating DTG than women initiating EFV. Overall, women initiating DTG were more likely to gain excess weight than women initiating EFV. This indicates that pregnant women on an INSTI regimen may be more at risk for excess weight gain, which could have adverse effects on the developing fetal CNS.

### 8.3. Folate Levels

Folate is essential for fetal neural tube development in the first 4 weeks of human pregnancy [146]. Thus, maternal folate level is considered one of the best predictors for the risk of neural tube defects (NTDs) [24]. Folate is crucial for one-carbon metabolism, significantly affecting DNA synthesis and cell division, and the conversion of homocysteine (Hcy) to methionine. Therefore, a deficiency in folate can lead to a decrease in the methylation cycle and an increase in neurotoxic Hcy. Subsequently, folate deficiency during fetal growth can result in insufficient closure of the neural tube and affected epigenetic mechanisms [147,148]. Further, placental implantation and development may also be affected. Overall, it has been well accepted that folate supplementation protects against NTDs [149]. Therefore, women with pregnancy potential are encouraged to take folic acid supplementation [146,150]. Many countries have implemented folic acid fortification for women with childbearing potential [150]. Also, exposure to folic acid antagonists during the first trimester of pregnancy has been associated with an increased risk of congenital malformations [151]. Therefore, the relationship between INSTI-related NTDs and disruption of the folate pathway has been of interest.

After the report from the Botswana Tsepamo study indicated that DTG exposure at conception was associated with increased risk of fetal NTDs, maternal serum folate concentrations were examined from stored samples of the South African ADVANCE trial [152]. In this study, 1053 treatment-naïve participants were randomized to initiate (TAF/FTC and DTG), TDF/FTC and DTG, or TDF/FTC/EFV. This study compared mean serum folate level changes at weeks 0, 12 and 24 across study arms. At baseline, maternal serum folate concentrations were similar across treatment arms. However, serum folate levels increased over 12 weeks in the TAF/FTC and DTG arm (+4.0 ± 8.1 nmol/L), while folate levels decreased slightly in the TDF/FTC and DTG arm (−1.8 ± 8.9 nmol/L). A higher decrease in folate levels was observed in the TDF/FTC/EFV arm (−5.9 ± 8.1 nmol/L). Interestingly, women with TDF/FTC/EFV had significantly lower folate concentrations at both 12 and 24 weeks compared with the other DTG arms. Of 26 women who became pregnant on study before week 24, the same trend was observed, in which folate concentrations increased between baseline and 12 weeks in both DTG arms but decreased in the TDF/FTC/EFV arm.

Tests of whether DTG and other INSTIs were clinically relevant inhibitors of the three major folate transport pathways were examined [153]. Tested folate transport pathways were proton-coupled folate transporter (PCFT), reduced folate carrier (RFC), and folate receptor a (FRα)-mediated endocytosis. DTG and CAB inhibited FRα, but not at clinically relevant concentrations. BIC inhibited both PCFT and FRα, but inhibition was not at clinically relevant concentrations. EVG and RAL inhibited PCFT. However, only RAL inhibition of PCFT was noted as potentially clinically relevant at the highest clinical dose. Overall, this study concluded that DTG is not a clinical inhibitor of folate transport pathways, and it is not predicted to elicit clinical decreases in maternal and fetal folate levels. Furthermore, a clinically relevant HIV integrase inhibitor drug class effect on folate transport pathways was not observed. In another study, DTG administered at therapeutic doses was associated with an increased risk of fetal NTDs in mice [25]. However, this study did not find any cause-and-effect relationship between DTG and folate and reported that DTG is unlikely to be an inhibitor of folate uptake. The same group further investigated the effect of DTG exposure on the functional expression of the RFC, PCFT, and FRα in human placental cell lines, human placental explants, and pregnant mice [154]. In placental cells, clinically relevant DTG exposure was associated with a significant reduction in the expression of RFC and PCFT, and uptake of RFC and PCFT substrates [^3^H]-methotrexate and [^3^H]-folic acid was noted to be decreased. Moreover, the author found in pregnant mice that DTG was associated with an increase in placental RFC and PCFT mRNA expression, accompanied by a decrease in placental FRα mRNA under folate-deficient dietary conditions. These data demonstrated a potential interaction between DTG and folate transport pathways in the placenta, potentially impacting folate delivery to the fetus under folate deficient conditions. Interestingly, a study conducted in zebrafish showed developmental toxicity after early embryonic exposure to DTG and reported that DTG is a partial antagonist of the folate receptor (FOLR1). Furthermore, the author reported that such DTG-associated developmental toxicity in zebrafish can be mitigated with supplementation of folic acid, signifying an effect of DTG on the folate receptor during early pregnancy [24].

### 8.4. Neurotoxicity

Clinical studies have showed high levels of transplacental transfer of INSTIs from mother to fetus during pregnancy and confirmed accumulation and prolonged elimination time from infants following gestational transfer (Table 2). Efficient levels of transfer of INSTIs from mother to fetus can potentially protect it from HIV-1 infection. However, at the same time, the fetus is exposed to drugs during a highly critical period of neurodevelopment, which may lead to drug-induced neurodevelopmental impairments. Our recent work in pregnant mice showed that DTG readily crosses the placenta and reaches the fetal CNS during gestation. Moreover, postnatal evaluation of brain health of pups which were exposed to DTG in utero showed neuronal injury and neuroinflammation during early postnatal development [23]. Our work demonstrated a potential direct association between in utero DTG transfer and developmental neurotoxicity. Currently, no additional reports have assessed DTG or other INSTIs effects on fetal functional neurodevelopment following in utero exposure using animal model. However, few researchers have evaluated the effects of individual INSTIs on cell culture systems or the CNS of adult rodents. EVG was found to be neurotoxic in primary rat neuroglia cultures as evidenced by a loss of microtubule-associated protein 2 [155]. EVG also was found to impair oligodendrocyte maturation and myelination [156]. In adult mice, treatment with EVG reduced remyelination, which was supported by in vitro studies of primary rat oligodendrocytes. Here, EVG inhibited oligodendrocyte maturation in a dose dependent manner and activated the integrated stress response (ISR). Latronico T et al. evaluated effects of ARVs in cultured primary astrocytes and showed that RAL induces oxidative stress [157]. Our research group studied the impact of DTG on the nervous system of adult mice using metabolomics. This study showed that DTG upregulated metabolomic markers of oxidative stress in different brain regions. In the same study, authors reported attenuation of DTG-induced metabolic oxidative stress when DTG was delivered as a poloxamer-based nanoformulation, signifying the need for development of drug delivery schemes to increase the efficacy and minimize toxicity associated during pregnancy [136]. There are very few pre-clinical studies which have evaluated direct effects of INSTIs on fetal neurodevelopment following usage during pregnancy or studied patho-mechanisms underlying potential adverse events. Thus, further research efforts are required to identify unknown adverse events on the developing brain and understand underlying mechanisms.

### 8.5. Bilirubin Neurotoxicity

ARVs that are primarily metabolized by UGT1A1 and have high protein binding properties, can potentially affect bilirubin metabolism, and compete with its albumin binding. This can potentially lead to hyperbilirubinemia in neonates [53,158]. Such hyperbilirubinemia may cause bilirubin associated central nervous system toxicity in neonates due to permeability of the blood–brain barrier at early age, especially in preterm neonates. Activity of UGT1A1 is low at birth and in the first few weeks of life [159]. Bilirubin is primarily metabolized by UGT1A1 [53], and so is RAL [54]. Moreover, RAL and bilirubin also compete for plasma albumin binding sites, which may lead to increased levels of unbound bilirubin in circulation. Such increased levels of unbound bilirubin likely cross the blood-brain barrier and cause bilirubin associated developmental neurotoxicity, clinically defined as kernicterus and/or bilirubin-induced neurologic dysfunction (BIND), leading to developmental neuroimpairments [53,160]. In a study determining PK of RAL in neonates following gestational exposure, one of twenty-two neonates received phototherapy treatment for hyperbilirubinemia [53]. Therefore, caution should be taken against high RAL concentrations in the neonate, as excessive levels may increase the risk of bilirubin neurotoxicity [53].

Recently, a case report evaluated the effects of initiation of RAL-based postnatal HIV-1 prophylaxis following transplacental exposure to DTG on neonatal bilirubin levels following birth [158]. Last DTG dose of mother was at 14 h before birth and prophylaxis treatment of the neonate with a RAL-based regimen was initiated 3.5 h after birth. RAL, DTG and bilirubin are primarily metabolized by the UGT1A1 pathway, and these metabolizing enzymes are immature at birth. During this early period after birth, RAL, DTG and bilirubin can potentially compete for metabolism by the UGT1A1 pathway and albumin binding. Thus, simultaneous high concentrations of RAL and DTG in neonates potentially increases the risk of hyperbilirubinemia and thus, bilirubin associated toxicity. Total bilirubin level in the neonate at day 6 after birth was 9.9 mg/dL, which was above population means, but well below the phototherapy treatment threshold of 21 mg/dL, according to their local protocol. Further, the study also analyzed DTG levels in the neonate and reported a substantially extended wash out half-life of DTG following in utero exposure. It was about 140 h against an average of 32 h in neonates who did not receive RAL as postnatal prophylaxis. Such prolonged half-life of DTG may be due to immaturity of UGT1A1 and competition among drugs and bilirubin. Thus, this study suggested potentially reduced metabolism, and thus increased risk of RAL, DTG or bilirubin induced toxicity. Overall, authors proposed to delay RAL-based postnatal HIV-1 prophylaxis initiation in neonates for 24–48 h after birth, given the mother has received a DTG-based regimen within 2–24 h before delivery to prevent potential toxicity.

### 8.6. Inhibition of Matrix Metalloproteinase (MMPs) Activity

Recently, we reported a novel pathway for DTG-associated developmental neurotoxicity that follows in utero exposure. The identified pathway was DTG inhibition of matrix metalloproteinase (MMPs) activities [23]. MMPs are a class of Zn++ dependent metalloenzymes. The role of MMPs in normal neural development is of critical importance. MMPs expression is at high levels during early CNS development and decreases into adulthood [161,162,163,164,165,166,167]. Due to their proteolytic activities, MMPs are ubiquitously expressed during neural development, however, their expression has been majorly studied in the hippocampus, cortex and cerebellum [165,168,169,170]. MMPs regulate multiple processes of neurodevelopment including, but not limited to, neurogenesis, neuronal migration, myelination, axonal guidance, synaptogenesis, synaptic plasticity, angiogenesis, and tissue remodeling [165,169,170,171,172,173,174,175,176,177,178,179]. Dysregulation of MMPs activity during critical periods of neurodevelopment can cause detrimental effect on neurodevelopmental mechanisms and may lead to neurodevelopmental disorders [165,169,170,171,172,173,174,175,176,177,178,179]. Interestingly, DTG possess a metal-binding pharmacophore (MBP) for engagement with active metal ion (Mg++) sites in the HIV-1 integrase enzyme to block its action [14]. With a metal chelating motif in the chemical structure, DTG possesses potential to interact with other metalloenzymes which are critical for normal biological activities. Thus, we hypothesized that DTG inhibition of MMPs activities during gestation can impair neurodevelopment [23]. We showed that DTG is a broad-spectrum MMPs inhibitor. It binds to Zn++ at the enzyme’s catalytic domain to inhibit activity. Moreover, studies in pregnant mice showed that DTG readily crosses the placenta and reaches to the fetal CNS during the critical period of brain development. Further postnatal evaluation of brain health in pups following gestational DTG exposure showed neuronal injury and neuroinflammation. In addition to DTG, all other INSTIs including CAB, BIC, RAL and EVG possess MBP to interact with metal ions [14]. Our preliminary molecular docking evaluation showed that CAB, BIC and RAL can interact with Zn++ at the catalytic domain of two MMPs 2 and 14. This preliminary assessment suggested that not only DTG, but the entire INSTI class possesses chemical abilities for broad-spectrum inhibition of MMPs activity.

Our work demonstrated an association between DTG dysregulation of MMPs activity during gestation and consequent postnatal neurotoxicity. Moreover, we showed potential linkage to the INSTI class effect. MMPs have multiple functions during CNS development and dysregulation of MMPs has been associated with impaired learning and memory, and neuropsychiatric disorders [168]. MMPs activity has been repeatedly demonstrated to be essential in influencing hippocampus-dependent learning and memory during development [168]. Studies assessing MMP knockouts, chemical inhibitors of MMPs, or animals with overexpression of MMPs reported impaired learning and memory on behavioral assessments and related long-term potentiation (LTP) in hippocampal pathways, confirming necessity of precisely regulated MMPs activity during development [165,168,180]. Thus, additional research investigations and continuous observations in humans are warranted to determine the causal relationship between INSTIs inhibition of MMPs during gestation and developmental neurotoxicity and identify unknown developmental deficits.

## 9. Long-Acting ARV Formulations and Neuroprotection

Our group recently demonstrated neuroprotective activities of long-acting (LA) poloxamer-coated DTG nanoparticles in adult mice [136]. We identified that DTG associated metabolomic dysfunctions in the brain were ameliorated when DTG was injected as a LA nanoformulation. Such a protective effect was considered due to neuroprotective properties of poloxamers and pharmacological properties of LA formulations. Poloxamers are known to have neuroprotective, anti-inflammatory, antioxidant properties; and poloxamers are also known to promote resealing of disrupted plasma membranes [181,182,183,184,185,186,187,188]. Moreover, sustained release of DTG from formulations and from tissue drug depots compared to high levels of DTG exposure from repeated daily administration of native drug would provide additional benefits [136,189]. Also, reduction in the total amount of drug and increase in the dosing interval, due to an increase in half-life, limits drug exposure compared to daily administration. For example, every day oral administration of clinically approved CAB (VOCABRIA) at 30 mg dose will end up with total 1680 mg of drug intake for 8 weeks against one intramuscular injection of CAB LA (CABENUVA or APRETUDE) of total 600 mg of drug every 8 weeks. Approximately 3-fold lower total drug administration compared to oral route due to PK properties of LA nanoformulations. Additionally, our group has developed DTG nanoformulations that can maintain clinically relevant therapeutic drug levels in mice for around one year following single intramuscular (IM) injection at a dosage of 45 mg/kg mouse weight [189,190]. Thus, such single injection of DTG LA nanoformulations will lead to several fold lower total drug administration than oral administration. Toxicology studies have shown that poloxamers are safe even at very high concentrations and have been approved by the US FDA for several pharmaceutical products [191]. Usage of LA formulations of ARVs in clinics established neuroprotective properties and safety of poloxamers, and reduction in drug exposure due to an increase in half-life and sustained release from depots, provides a novel means to provide therapeutic benefits for DTG during neurodevelopment. Moreover, these formulations achieve therapeutic levels in mother to control viral infection [73,189,192] and can also help to prevent known adverse effects of DTG on the mother [86], improving maternal health during pregnancy. Utilization of nanoformulations to increase efficacy and safety of drugs in other diseases during pregnancy provides validation of such concept [193,194]. With CAB LA being introduced in clinics [71,72], and utilization of polymer surfactants for ARV LA [73,192], development of LA formulations of other INSTIs for therapeutic benefits during pregnancy provides novel potential means for translational preventive strategies.

## 10. Conclusions and Future Directions

Worldwide, over one million HIV-1-infected pregnant women give birth each year. Due to current recommendations for ARVs usage during pregnancy, the majority of these children are and will be exposed to INSTIs, particularly in RLCs. Observational clinical and pre-clinical studies reported increased risks of NTDs and postnatal neurodevelopmental impairments following in utero DTG exposure. These adverse effects were prominent following DTG usage at conception. To this date, underlying mechanisms or a link to INSTI class effect is unknown. In full consideration of prior human studies, it is timely to elucidate any potential links to adverse events, no matter how infrequent, in order to provide the most effective care to pregnant women and their fetuses. Therefore, it is essential to identify unknown adverse effects and address linked potential mechanisms underlying compromised neurodevelopmental outcomes. Continued clinical monitoring and more basic science research are critically needed to understand clinical outcomes and linked mechanistic pathways. These can help to develop prevention and intervention strategies. Finally, further understanding for long-acting drug formulation as a safe delivery system is needed, which could potentially serve to maximize the drug’s benefits and minimize any untoward effects.

## Figures and Tables

**Figure 1 pharmaceuticals-15-01533-f001:**
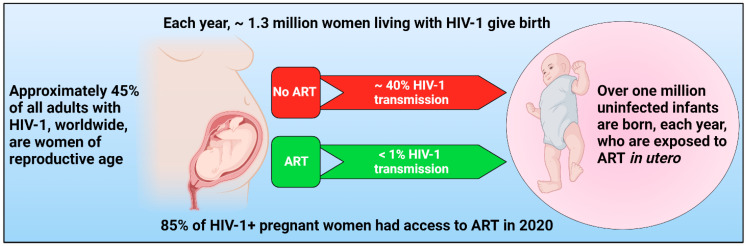
**Antiretroviral therapy and mother-to-child HIV-1 transmission.** Currently, around 45% of all adults living with HIV-1 infection are women of child-bearing age. Every year, around 1.3 million HIV-1 infected women give birth. Due to increased accessibility of affordable antiretroviral therapy in both resource-rich and resource-limited countries (RRCs and RLCs), vertical transmission rate of HIV-1 infection from mother-to-child has significantly reduced from around 40% to less than 1% with high adherence and accessibility to antiretroviral therapy (ART).

**Figure 2 pharmaceuticals-15-01533-f002:**
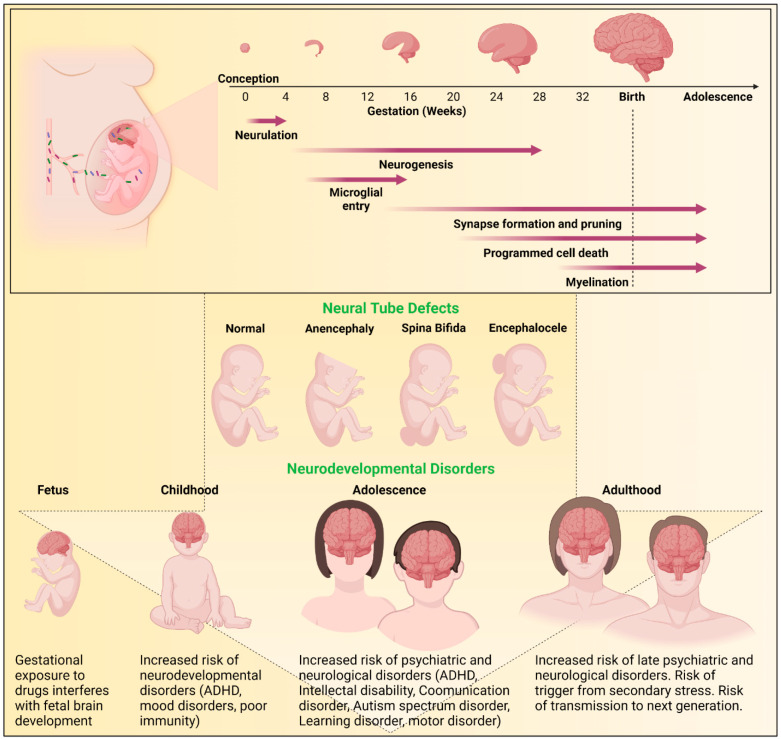
**Outline of brain development and risks of neurodevelopmental disorders.** The human brain undergoes critical stages of development starting from conception. Neurulation involves the closure of the neural tube. Neurogenesis involves the differentiation of neural stem cells into neurons. Microglial entry occurs when microglia invade the central nervous system (CNS) to establish a resident immune cell population. Synaptogenesis and the pruning of excess synapses and dendritic spines occurs to establish neural circuits. Cell populations are further refined through apoptosis. Finally, myelination of neurons occurs. While some processes occur only during gestation, others continue into childhood and adulthood. These neurodevelopmental processes are highly susceptible to external modulators such as drugs, virus, and inflammatory proteins. Any insults to the neurodevelopmental biological, metabolic, or physiological processes can have serious adverse effects, including neural tube defects (NTDs) or postnatal neurodevelopmental disorders.

**Figure 3 pharmaceuticals-15-01533-f003:**
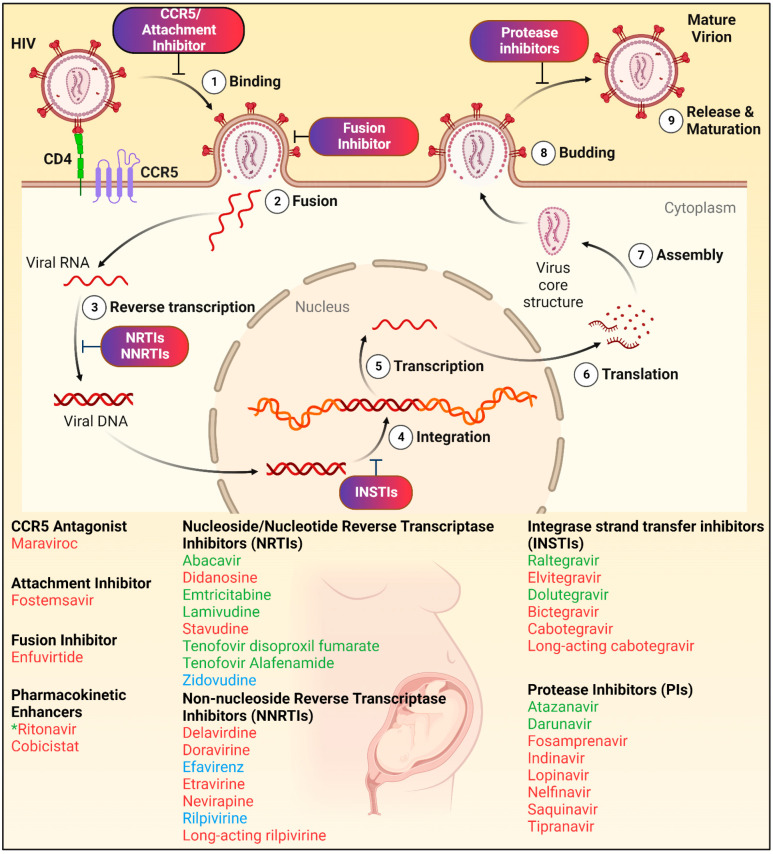
**Overview of HIV-1 life cycle and antiretroviral drugs during pregnancy.** HIV-1 life cycle is classified into nine major steps include binding, fusion, reverse transcription, integration, transcription, translation, assembly, budding and maturation. Each step of the HIV-1 life cycle is a potential target for antiretroviral drugs (ARVs). A list of current the recommendations for use of ARVs for infected pregnant women who have never received ARVs before (ARV-naïve), set by the United States Department of Health and Human Services (DHHS) Panel on Treatment of HIV During Pregnancy and Prevention of Perinatal Transmission, is provided. Recommended categorization of individual ARVs is as *preferred*, *alternative*, or *not recommended*. *Preferred* ARVs: highlighted in green color, *Alternative* ARVs: highlighted in blue color, and *Not recommended* ARVs: highlighted in red color. * Note: Ritonavir (RTV) is only *preferred* when used as a pharmacokinetic (PK) booster for PIs include darunavir (DRV) and atazanavir (ATV) during pregnancy.

**Table 1 pharmaceuticals-15-01533-t001:** Antiretrovirals (ARVs) and recommendations for their usage during pregnancy.

Class	Drug name	Acronyms	US FDA Approval Year	Recommendations for Pregnant Women Who Are ART Naïve	Recommendations for Continuation of ART in Women Who Become Pregnant	Recommendations for Women Planning to Become Pregnant
	Tenofovir Alafenamide	TAF	2016	Preferred	Continue	Preferred
**Nucleoside Reverse Transcriptase Inhibitors (NRTIs)**	Emtricitabine Tenofovir disoproxil fumarate Abacavir Lamivudine	FTC TDF ABC 3TC	2003 2001 1998 1995	Preferred Preferred Preferred Preferred	Continue Continue Continue Continue	Preferred Preferred Preferred Preferred
	Zidovudine	ZDV	1987	Alternative	Continue	Alternative
	Long-acting rilpivirine	LA-RPV	2021	Not recommended	Insufficient data	Insufficient data
	Doravirine	DOR	2018	Insufficient data	Insufficient data	Insufficient data
**Non-Nucleoside Reverse Transcriptase Inhibitors (NNRTIs)**	Rilpivirine Etravirine Efavirenz	RPV ETR EFV	2011 2008 1998	Alternative Not recommended Alternative	Continue Continue Continue	Alternative Not recommended, except in special circumstances Alternative
	Nevirapine	NVP	1996	Not recommended	Continue	Not recommended, except in special circumstances
	Darunavir	DRV	2006	Preferred	Continue	Preferred
	Tipranavir	TPV	2005	Do not use	Switch	Do not use
	Fosamprenavir	FPV	2003	Do not use	Switch	Do not use
**Protease Inhibitors (PIs)**	Atazanavir	ATV	2003	Preferred	Continue	Preferred
	Lopinavir	LPV	2000	Not recommended, except in special circumstances	Continue	Not recommended, except in special circumstances
	Saquinavir	SQV	1995	Do not use	Switch	Do not use
**Fusion Inhibitor**	Enfuviritide	T-20	2003	Not recommended	Continue	Not recommended, except in special circumstances
**CCR5 Antagonist**	Maraviroc	MVC	2007	Not recommended	Continue	Not recommended, except in special circumstances
	Cabotegravir	CAB	2021	Not recommended	Insufficient data	Insufficient data
	Long-acting cabotegravir	LA-CAB	2021	Not recommended	Insufficient data	Insufficient data
**Integrase Strand Transfer**	Bictegravir	BIC	2018	Insufficient data	Insufficient data	Insufficient data
**Inhibitors (INSTIs)**	Elvitegravir	EVG	2014	Not recommended	Continue with viral monitoring or consider switching	Not recommended
	Dolutegravir	DTG	2013	Preferred	Continue	Preferred
	Raltegravir	RAL	2007	Preferred	Continue	Preferred
**Attachment Inhibitor**	Fostemsavir	FTR	2020	Insufficient data	Insufficient data	Insufficient data
**Pharmacokinetic Enhancers**	Cobicistat	COBI, c	2014	Not recommended	Not recommended	Not recommended
	Ritonavir	RTV	1996	Depends on Protease Inhibitor	Depends on Protease Inhibitor	Depends on Protease Inhibitor

**Table 2 pharmaceuticals-15-01533-t002:** Overview of PK of INSTIs during pregnancy.

	Raltegravir			Elvitegravir/Cobicistat			Dolutegravir			Bictegravir
**Route of administration**	Oral			Oral			Oral			Oral
**Dosing**	400 mg twice daily			150/150 mg once daily			50 mg once daily			50 mg once daily
**Metabolism**	Primarily UGT1A1			Primarily CYP3A4 Minor UGT1A1			Primarily UGT1A1 Minor CYP3A4			Equal contribution by CYP3A4 and UGT1A1
**Protein Binding**	76–83%			Elvitegravir 98–99%			Approximately 99%			Approximately 99.7%
**AUC0-12 or 0–24 (µg*h/mL)**	[43] ^ad^ 2^nd^ trimester: 6.6 (2.1–18.5) 3^rd^ trimester: 5.4 (1.4–35.6) Postpartum: 11.6 (1.6–39.9)	[44] ^bd^ 3^rd^ trimester: 5.0 (3.56–7.01) Postpartum: 7.11 (4.91–10.30)	[45] ^ad^ 3^rd^ trimester: 3.921 (0.699–14.706) Postpartum: 6.770 (1.506–21.859)	[46] ^ae^ 2^nd^ trimester: 15.283 (11.939–19.038) 3^rd^ trimester: 14.004 (9.119–18.798) Postpartum: 21.039 (13.532–32.788)	[47] ^ace^ 3^rd^ trimester: 14.1 (39) Postpartum: 21.7 (29)	[48] ^f^ 3^rd^ trimester: 14.339 Postpartum: 15.356	[49] ^ae^ 2^nd^ trimester: 47.6 (33.4 – 63.7) 3^rd^ trimester: 49.2 (36.4 –62.0) Postpartum: 65.0 (47.8–88.4)	[50] ^be^ 3^rd^ trimester: 35.322 (19.196–67.922) Postpartum: 40.127 (22.795–59.633)	[51] ^ace^ 3^rd^ trimester: 40.8 (35) Postpartum: 47.0 (42)	[52] ^e^ 3^rd^ trimester: 37.9 Postpartum: 58
**Cmax (µg/mL)**	2^nd^ trimester: 2.250 (0.365–5.960) 3^rd^ trimester: 1.770 (0.315–7.820) Postpartum: 3.035 (0.312–12.600)	3^rd^ trimester: 1.43 (0.93–2.22) Postpartum: 1.76 (1.10–2.80)	3^rd^ trimester: NR Postpartum: NR	2^nd^ trimester: 1.447 (1.133–1.579) 3^rd^ trimester: 1.432 (0.705–1.570) Postpartum: 1.713 (0.955–2.284)	3^rd^ trimester: 1.4 (42) Postpartum: 1.8 (26)	3^rd^ trimester: 1.270 Postpartum: 1.352	2^nd^ trimester: 3.62 (2.57–4.63) 3^rd^ trimester: 3.54 (2.66–4.24) Postpartum: 4.85 (3.83–5.97)	3^rd^ trimester: 2.534 (1.462–3.986) Postpartum: 2.899 (1.397–4.224)	3^rd^ trimester: 3.15 (31) Postpartum: 3.34 (32)	3^rd^ trimester: 3.82 Postpartum: 4.7
**Ch or trough (µg/mL)**	2^nd^ trimester: 0.0621 (0.0128–0.438) 3^rd^ trimester: 0.064 (0.0114–0.607) Postpartum: 0.0797 (0.0199–1.340)	3^rd^ trimester: 0.077 (0.043–0.137) Postpartum: 0.120 (0.074–0.193)	3^rd^ trimester: 0.057 (0.003–0.277) Postpartum: 0.069 (0.005–0.248)	2^nd^ trimester: 0.025 (0.017–0.067) 3^rd^ trimester: 0.048 (0.014–0.075) Postpartum: 0.377 (0.228.5–0.568)	3^rd^ trimester: NR Postpartum: NR	3^rd^ trimester: 0.018 Postpartum: 0.068	2^nd^ trimester: 0.73 (0.63–1.34) 3^rd^ trimester: 0.93 (0.68–1.34) Postpartum: 1.28 (0.80–1.95)	3^rd^ trimester: 0.642 (0.188–0.3088) Postpartum: 0.777 (0.348–0.1210)	3^rd^ trimester: 0.68 (84) Postpartum: 1.03 (68)	3^rd^ trimester: 0.63 Postpartum: 1.23
**Tmax (h)**	2^nd^ trimester: 4.0 (1.0–8.0) 3^rd^ trimester: 2.0 (0–12.0) Postpartum: 2.0 (0–8.0)	3^rd^ trimester: 1.98 (0–11.3) Postpartum: 2.03 (0–7.97)	3^rd^ trimester: NR Postpartum: NR	2^nd^ trimester: 4 (2–4) 3^rd^ trimester: 4 (2–6) Postpartum: 4 (2–6)	3^rd^ trimester: 4 (2.0–6.0) Postpartum: 5 (3.0–7.8)	3^rd^ trimester: 2.9 Postpartum: 3.0	2^nd^ trimester: 2 (2–4) 3^rd^ trimester: 3 (2–4) Postpartum: 2 (2–4)	3^rd^ trimester: NR Postpartum: NR	3^rd^ trimester: 3.0 (1.0–4.5) Postpartum: 3.8 (0.5–8.0)	3^rd^ trimester: NR Postpartum: NR
**T1/2 (h)**	2^nd^ trimester: 2.9 (1.2–85.6) 3^rd^ trimester: 3.7 (1.1–211.7) Postpartum: 3.6 (1.1–30.5)	3^rd^ trimester: 2.55 (1.88–3.45) Postpartum: 2.53 (1.91–3.36)	3^rd^ trimester: NR Postpartum: NR	2^nd^ trimester: 3.1 (2.6–3.9) 3^rd^ trimester: 3.4 (2.7–4.7) Postpartum: 8.8 (7.0–13.2)	3^rd^ trimester: 4.3 (37) Postpartum: 7.7 (30)	3^rd^ trimester: 2.6 Postpartum: 3.7	2^nd^ trimester: 11.0 (8.9–13.1) 3^rd^ trimester: 12.2 (10.4–15.0) Postpartum: 13.5 (10.6–18.6)	3^rd^ trimester: NR Postpartum: NR	3^rd^ trimester: 10.5 (49) Postpartum: 14.4 (46)	3^rd^ trimester: NR Postpartum: NR
**Cord blood:maternal blood ratio**	1.5	1.21	NR	0.91	0.75	0.64	1.25	1.21	1.29	1.49
**Placental Transfer**	High			High			High			High
**T1/2 (h) in infants post delivery**	26.6 (9.3–184) ^a^ [53]			7.6 (6.3–10.2) ^a^ [46]			32.8 (25.9–35.9) ^a^ [49]			NR
**Dosing recommendations in pregnancy**	No change in dose indicated			Insufficient data to make dosing recommendations			No change in dose indicated			Insufficient data to make dosing recommendations

^a^ Median (IQR: interquartile range); ^b^ Geometric mean with 95% confidence interval (CIs); ^c^ Geometric mean (coefficient of variation, %); ^d^ AUC_0-12_ (µg*h/mL), C_12h_ (µg/mL), or C_trough_ (µg/mL); ^e^ AUC^0−24^ (µg*h/mL), C_24h_ (µg/mL), or C_trough_ (µg/mL); ^f^ AUC_0–24_ (µg*h/mL), C_0h_ (µg/mL), or C_trough_ (µg/mL); AUC = area under the plasma concentration; C_0h_ = predose concentration; C_12h_ = concentration 12 h after last dose; C_24h_ = concentration 24 h after last dose; C_max_ = maximum concentration; T_max_ = time to reach maximum concentration post- dose; T_1/2_ = half-life; NR: not reported.

## Data Availability

Data sharing not applicable.

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
