# Peer review of "HIV-1 Integrase Strand Transfer Inhibitors and Neurodevelopment"

_pharmaceuticals, 2022, doi:10.3390/ph15121533_

Round 1
Reviewer 1 Report
Dear authors, congratulations for the work done writing the manuscript with title HIV-1 Integrase Strand Transfer Inhibitors and Neurodevelopment.
It is a well done review. The matter is actual and important, children born to mothers with or at risk of human immunodeficiency virus type-1 (HIV-1) infection and exposed to antiretroviral drugs during pregnancy, and the natal neurodevelopment incompletely understood. The periconceptional usage of dolutegravir (DTG) pose the potential risks of neural tube defects (NTDs). Incidence studies of neurodevelopmental outcomes associated with DTG, and other integrase strand transfer inhibitors (INSTIs) are limited as widespread use of INSTIs has begun only recently in pregnant women. Therefore, any associations between INSTI use during pregnancy, and neurodevelopmental abnormalities remain to be explored. Updates on INSTI pharmacokinetics and adverse events during pregnancy together with underlying mechanisms which could affect fetal neurodevelopment are also discussed. Overall, the review brings together clinical and basic scientists information on potential consequences of INSTIs on fetal outcomes serving future scientific investigations.
Reviewer 2 Report
The authors report in this review updates on INSTI pharmacokinetics and adverse events during pregnancy together with underlying mechanisms which could affect fetal neurodevelopment. The review is well written, covering the United States Food and Drug Administration approved ARVs and their use during pregnancy, and seeking to educate both clinical and basic scientists on potential consequences of INSTIs on fetal outcomes serving future scientific investigations. Overall, the work is solid and interesting, there are a few concerns that I hope the authors can address. This manuscript may be considered for publication in the Pharmaceuticals after modification.
Minor concerns:
1. In part of Antiretroviral Drugs (ARVs), the authors describe different classes ARVs, entry inhibitors, fusion inhibitors, NRTIs, NNRTIs, INSTIs, protease inhibitors, and PK enhancers, and that can be supplemented with the addition of RNase H inhibitors, capsid modulators, transcriptional inhibitors, auxiliary protein inhibitors, etc.
2. In part of INSTI PK Profiles During Pregnancy, please discuss what physiological changes during pregnancy that drastically reduce drug levels to be below therapeutic requirements and leading to virologic failure.
3. In part of Weight Gain of INSTI Adverse Events, “With ART initiation, viremia and the associated inflammation subside, leading to weight gain.” “This has been found both in adults and children switching to DTG.” Are there studies discussing whether this adverse event of weight gain occurs in fetuses being exposed to DTG in utero.
4. The figure notes are too much and can be described in the main text.
